# Inventive Microstructural and Durability Investigation of Cementitious Composites Involving Crystalline Waterproofing Admixtures and Portland Limestone Cement

**DOI:** 10.3390/ma13061425

**Published:** 2020-03-20

**Authors:** Pejman Azarsa, Rishi Gupta, Alireza Biparva

**Affiliations:** 1Department of Civil Engineering, Facility for Innovative Materials and Infrastructure Monitoring (FIMIM), University of Victoria, 3800 Finnerty Rd., Victoria, BC V8P 5C2, Canada; pazarsa@uvic.ca; 2Kryton International Inc., 1645 East Kent Ave N, Vancouver, BC V5P 2S8, Canada; alireza@kryton.com

**Keywords:** durability of cement-based materials, microstructural formation, smart construction materials, sustainability, permeability reducing admixtures (PRA), crystalline waterproofing admixtures (CWA)

## Abstract

The durability of a cement-based material is mainly dependent on its permeability. Modifications of porosity, pore-structure and pore-connectivity could have significant impacts on permeability improvement, which eventually leads to more durable materials. One of the most efficient solutions in this regard is to use permeability reducing admixtures (PRA). Among these admixtures for those structures exposed to hydro-static pressure, crystalline waterproofing admixtures (CWA) have been serving in the construction industries for decades and according to ACI 212—chemical admixtures’ report, it has proven its capability in permeability reduction and durability-enhancement. However, there is substantial research being done on its durability properties at the macro level but very limited information available regarding its microstructural features and chemical characteristics at the micro level. Hence, this paper presents one of the first reported attempts to characterize microstructural and chemical elements of hydration products for cementitious composites with CWA called K, P and X using Scanning Electron Microscopy (SEM). Backscattered SEM images taken from a polished-section of one CWA type—K—admixture were analyzed in ImageJ to obtain paste matrix porosity, indicating a lower value for the CWA-K mixture. X-ray analysis and SEM micrographs of polished sections were examined to identify chemical compositions based on atomic ratio plots and brightness differences in backscatter-SEM images. To detect chemical elements and the nature of formed crystals, the fractured surfaces of three different CWA mixtures were examined. Cementitious composites with K admixture indicated needle-like crystal formation—though different from ettringite; X and P admixtures showed sulfur peaks in Energy Dispersive Spectrum (EDS) spectra, like ettringite. SEM images and X-ray analyses of mixtures incorporating Portland Limestone Cement (PLC) indicated lower-than-average porosity but showed different Si/Ca and Al/Ca atomic ratios.

## 1. Introduction

The porous nature of cementitious composites can be one of the main sources of their proneness to degradation. Due to such porous structures, aggressive agents (e.g., chloride ions) can be carried by water and penetrate through pores and micro-cracks, which endangers the composite’s microstructure, thereby leading to a reduction in its overall durability. In the aforementioned framework, there is an increasing interest in innovative and sustainable approaches to inherently reduce the permeability of cementitious materials as a promising, direct way to improve its durability at different scales. In real applications for cementitious materials, chemical admixtures are widely used [1,2,3,4] to enhance the physical, mechanical and durability properties of cementitious materials. Among these chemical admixtures are permeability reducing admixtures (PRA), which are recommended by the American Concrete Institute (ACI) committee 212 [5] and have been used by construction industries for decades, to fulfill such purposes. One of the innovative materials in the PRA category is the crystalline waterproofing admixture (CWA).

As a smart construction material, crystalline products have been used for over three decades in the construction industry and their effects on concrete self-healing, durability and shrinkage behaviour have been investigated in some respects. The visual closure of cracks in mortar specimens incorporating fly ash, expansive admixtures, silica fume, CWA and limestone powder under water immersion conditions was investigated by Jaroenratanapirom and Sahamitmongkol [6,7]. Self-healing (SH) potential and water tightness of pre-cracked cement-based materials treated with calcium sulfoaluminate (CSA)-based expansive additive and CWA were also explored in Sisomphon et al.’s study [8]. They also examined the healing effects of CWA on strength recovery under four different exposure conditions (wet/dry cycles, humidity chamber, water immersion with/without renovation and air exposure) [9]. Ferrara et al. [10] studied the CWA effects on the concrete SH and their healing capabilities toward the recovery of mechanical properties; they evaluated the influences of the SH phenomena on the recovery of stiffness and load-bearing capacity by means of 3-point bending (3pb) test before and after conditioning [10]. Using a different technique, a similar study by Roig-Flores et al. [11] investigated the effects of CWA only on the SH of concrete in four various types of environmental exposure conditions. Based on the measure of the global permeability of the specimen and different geometrical characteristics of the crack before and after the SH period, they developed a method that can evaluate the SH properties of cracked samples. Following the previous study, Roig-Flores et al.’s [12] work analyzed the SH properties of early-age concretes, engineered using CWA, by measuring the permeability of cracked samples and their crack width. Under three different exposure conditions, they considered the SH behavior in two typically used concrete classes, one common for precast concrete elements (C45/55) and one standard class broadly used for building construction (C30/37) [12]. In a recent study by Ferrara et al. [13], the influences of CWA on the self-healing capacities of the cementitious composites with reference to both a normal strength concrete (NSC) and a high-performance fiber reinforced cementitious composite (HPFRCC) have been evaluated. Pazderka and Hájková study [14] reported the results of the laboratory testing of the speed of the waterproofing effect, caused by the crystalline waterproofing admixture in concrete; they have shown that the concrete structure with CWA could be (theoretically) ready for water loading on the 12th day after creation. The laboratory measurement has indicated that the CWA reduces the water vapor permeability of concrete up to 20% [14]. Drochytka et al.’s [15] work demonstrates that the polymer cement waterproofing materials with CWA can be successfully modified with fly ash additive in order to reduce the content of cement in the formulations by 10%. It was also observed that the needle-shaped crystals with the maximum length of 15 μm of single crystals mostly filled out pores with sizes of about 20 μm [15]. Žižková et al.’s [16] indicated that the best conditions for storing mortars containing CA in terms of optimal porosity are 95% relative humidity and a temperature of 23 °C and also the addition of CA (added in an amount of 1.5% of cement mass) improves the 28-day compressive strength of the mortars. In another study, Žižková et al. [17] reported an improvement in the resistance of cement-based mortars against cyclic freezing as well as gaseous CO_2_. An important parameter that influences the effectiveness of the CWA is the curing conditions during the first 28 days of ageing. They also concluded that CWA has an influence on the increase of the amount of portlandite (CH) [18]. SEM sampling revealed differences in the microstructures of mortars both with and without CWA after exposure to aggressive environments [18]. SEM observations and EDS analyses of Cuenca et al.’s study [19] confirmed the presence of healing products on the healed surfaces mainly due to the delayed hydration and carbonation reactions involving both the cement and the crystalline waterproofing admixture. The morphologies of these products and their relative EDS analyses also suggest that some ettringite crystals (hydrous calcium aluminum sulfate mineral) may form inside and fill the same cracks. It was proven in Nevřivová et al. [20] that the addition of polypropylene (PP) fibers increases the apparent porosity of the mortar by 10% to 15%, and if the mortar contains a CWA, the fibers increase the porosity by up to 50%. The CWA reduces the mortar’s apparent porosity. For larger crack widths, Abro et al. [21] showed that the recovery of the diffusion coefficients depended on the mixture. The mixture incorporating calcium sulfoaluminate as an expansive admixture and bentonite as a swelling agent together with a CWA including Na_2_CO_3_ and organic calcium ions showed better self-healing performance than the mixture incorporating only CWA. Li et al.’s [22] study demonstrated that the crack healing efficiency of Super Absorbent Polymer (SAP)-added specimen was quite satisfactory, but the complete repair was hard to achieve. The healing capacity of CWA for micro-crack is good; however, a crack more than 0.3 mm is hard to be sealed. Hence, the combination of SAP and CWA was recommended for macro-crack sealing. Elsalamawy et al.’s [23] study indicated that different phases of calcium silicate hydrate C–S–H have been formed in presence of CWA materials. These phases range from round particles to needle-like crystals with Ca/Si ratio ranging from 2.4 to 3.2. It was also observed that the higher the (w/c) ratios, the higher the efficiency of the using “CWA” materials regarding the enhancement to the water capillary’s resistivity.

To better understand the influence of innovative materials like CWA in terms of improving durability and strength of concrete, it is essential to first obtain information about CWA’s microstructure which provides useful physical and chemical characteristics of the material. Microstructural examination of such new construction materials can also lead to better implementation and development of these materials for the construction and building industries. CWA has been utilized for (decades) in different arenas of the construction industry; only a few recent studies have been identified that address the effect of CWA in the cement-based materials as a promoter of permeability and self-healing [6,7,8,9,10,11,12,13,14,15,16,17,18,19,20,21,22,23,24,25,26,27,28,29,30,31,32,33,34,35,36]. However, none of these studies have attempted to understand the development and behavior of CWA at the micro level which has a direct influence on its macro mechanical/physical properties. Additionally, the interaction of CWA and Portland limestone cement (PLC) in the microstructure of a cementitious composite is not known and is yet to be confirmed. In addition, differences in chemical composition and the morphological properties of various CWA types have not been reported in any studies.

This paper aims to investigate the role of the addition of different CWA and PLC on the development of microstructure, nature of hydration phases, porosity reduction of paste matrix and morphological properties of cementitious composite. This study has been organized into three different phases. Phase I covers detailed information about paste’s pore network, determined by implementing an image analysis technique on backscattered mode scanning electron microscopy (BSEM) images, taken from polished sections of cementitious composite mixtures containing only one CWA type (called K in this study) and Portland limestone cement (PLC). Phase II, using BSEM micrographs and the energy dispersive X-ray spectroscopy (EDS) technique, microstructural features, the development of pore-blocking crystals and the chemical components of similar cementitious composite mixtures are investigated. This phase was also designed to identify differences in hydration products and mineral formation while CWA was present in the mix. Although hydration products and morphology were successfully identified in Phase II, formation of crystals inside the matrix was not observed, mostly due to damage induced during harsh surface preparation required for conventional polished surface preparation method. Hence, the detrimental effect of surface preparation in Phase II led to finding an alternative surface preparation technique that supports crystal growth as claimed by CWA’s manufacturer. One of the main goals of this study was to demonstrate whether or not CWA form similar microstructures (pore-blocking crystals) or share the same chemical hydration products. In order to investigate the nature of pore-blocking crystals and their chemical compositions, Phase III was initiated, which delved into SEM and EDS results taken from fractured surfaces of cementitious composites treated with different CWA types (called K, P and X). For all treated CWA mixtures, the observed needle shape crystals were quite similar to ettringite crystals, indicating the need for further analysis of their EDS spectra. Chemical analysis of these crystals showed a high sulfur peak in the EDS graphs for X and P admixtures similar to the ettringite spectra, but this was not the case for K admixture. Ultimately, the last section in this paper draws final conclusions and summarizes the authors findings obtained from all phases. 

## 2. Experimental Methodology

### 2.1. Phase I 

A previous study by the authors [26] investigated the effects of a CWA admixture (K admixture in this case) on the permeability of concrete, using the DIN 1048 test method [37] to evaluate the effects of this product on permeability. Based on the permeability study, it is reported that K admixture can reduce the coefficient of permeability (COP) by more than 60%. Hence, this phase aims to conduct an image analysis technique at the micro level to understand the pore network and porosity of cement-based system treated with CWA; and later, to discover any correlation between the permeability and porosity of uncracked specimens.

Generally, a CWA consists of a proprietary mix of active chemicals, implanted in a carrier of cement and sand, reacting with cement compounds (K admixture) or by-products of cement hydration, most likely Ca(OH)_2_ (X and P admixtures) as described by the American Concrete Institute (ACI) Committee 212 [5]. In the study of Sisomphon et al. [8], it is stated that calcium hydroxide (CH) is the reactive component when tested for a type of CWA. As a result of the chemical reaction and deposition of integrally bonded crystals into the hardened cement paste, pressure resistance of modified matrix increases as high as 14 bars [5]. The chemical compositions of these products are proprietary and not available; however, some of the chemical and physical properties of the material, reported by its manufacturer and used in this study, are given in Table 1. In compliance with ASTM C150 [38], Quikrete Portland cement-type 10 (also referred as Type GU in CSA A23.1-14 [39]) was used for mortar mixtures in this study. Portland limestone cement (PLC) was also provided by Lafarge Inc., Vancouver, BC. Naturally available fine aggregate, meeting ASTM C33 [40] requirement, from Sechelt pit in B.C., Canada, was used for the experimental work. Moisture content was measured by the change in weight of a sample after keeping the sample for 24 h in an oven at 100 °C. Calculated moisture content was 2.15%. Fineness modulus, density and absorption of fine aggregate were 2.60, 2.65 and 0.79% respectively in accordance with ASTM C128 [41]. 

In the first phase, polished and flat surfaces of cementitious composite specimens were used to systematically analyze porosity of the treated system using BSEM micrographs and ImageJ software (2018), National Institutes of Health, Bethesda, Maryland, USA. To keep the scope of this project manageable and due to failure in detecting needle-like crystals after harsh surface preparation procedure, only one type of CWA (K admixture) and two cement types (Ordinary Portland Cement (OPC) and Portland Limestone Cement (PLC)) were used in this experiment. A typical dosage (2% weight of cement) was used in this experiment to understand actual behavior of CWA in blocking pores or cracks. Table 2 summarizes the mix designs, used for all phases with a water/binder ratio of 0.5 to achieve enough flow of water in the matrix. Initially, standard water curing age (28 days) has been considered for the first two phases but through lessons learned from experiments in these phases, longer curing age (56 days) was set for Phase III to ensure that specimens had enough time for fully-hydration process and crystal growth.

The mortar samples were prepared following the procedure reported in ASTM C305 [42]. First, all the mixing water was placed in the bowl. Then, cement was added to water and the mixer was started at the slow speed for 30 s. Later, the entire quantity of sand was slowly added over a 30-s period, while mixing at slow speed. After mixing in medium speed for 30 s, the mixer was stopped for 90 s to let the mortar stand. Finally, the mixing was continued for another 60 s before placing mortar into 50 × 50 × 50 mm^3^ cubic molds. Promptly after the casting samples, they were sealed with a plastic sheet to keep almost constant moisture inside for 24 h at laboratory environment (room temperature) during concrete stiffing and strength development. After 24 h of hardening and prior to testing, samples were continuously cured in a water bath at 23 ± 2 °C.

To prepare polished sections, three cubes (50 × 50 × 50 mm^3^) from each mix were cut into thin slices using a diamond saw so that edging is maintained. These 10 mm thick slices were then embedded in low viscosity epoxy resin using vacuum impregnation in a vacuum desiccator. Once the resin was hardened, a few millimeters from the surface were removed using wet-and-dry abrasive sheet to reveal surface to be examined. Later, the specimen surface was ground to a flat surface using a polishing/lapping machine. The time spent on each grinding and polishing stage was about 10–15 min. The flat surface was cleaned in acetone, dried and made ready for carbon coating. After applying carbon-coating by an evaporative coater, the coated sections were ready to be studied by Hitachi S-4800 SEM, Hitachi, Chiyoda City, Tokyo, Japan, with a backscatter electron detector and Energy Dispersive Spectrum (EDS) analysis. The microscope was operated at an accelerating voltage of 15 kV and working distance 15 mm. The EDS system had been calibrated with oxide standards suitable for cement analyses. X-ray spectra and elemental mappings were collected for elements O, Na, Mg, Al, Si, S, K, Ca, Ti, Mn and Fe, which are the typical cementitious materials’ hydration products [43].

#### Image Analysis of Polished Sections

The images were taken from random locations on the specimen to eliminate any operator bias, while attempts were made to not capture them from the cubes’ finished surface or edges to minimize areas that may have experienced contamination. Image analysis was performed using ImageJ software developed by the National Institutes of Health [44]. To draw statistically meaningful conclusions, a total of 10 locations on each sample were imaged at a magnification of 800×, which is a common magnification used to study concrete pore structures [45]. A representative unfiltered image (magnification: 800×) taken by backscatter mode of SEM from polished surface of L-K mix, is shown in Figure 1a. These images typically contain some fraction of sand particles which need to be removed from original BSEM image using a segmentation algorithm [46] before thresholding and segmentation of the pore phase. This is mainly due to overlap with hydrated cement paste grayscale. An initial step was to use filtering operations to emphasize certain feature and reduce noise in SEM micrograph (Figure 1b). The image processing step shown in Figure 1b, implemented with the filtering operations of smoothing, sharpening and edge enhancement. For cementitious composite specimens, where there was a pore peak in the histogram, the minimum point between the pore peak and the hardened paste peak was considered as the porosity threshold. This process was done by referring to the first derivation curve of the brightness histogram. In the ImageJ software, auto-threshold utility, an iterative procedure based on the isodata algorithm [47], was also used for thresholding. The procedure divides the image into object and background by taking an initial threshold; then the averages of the pixels at or below the threshold and pixels above are computed. The averages of those two values are computed, the threshold is incremented and the process is repeated until the threshold is larger than the composite average [48]. Adjustment of contrast and brightness was also performed to balance grayscale level (Figure 1c). When the pore thresholds had been determined, binary segmentation was conducted on the BSEM images to distinguish pores from other phases (Figure 1d). Based on a percentage of the segmented pore area to the total paste area, porosity of hardened paste was later calculated for 10 specimens for each mixture. Additionally, since EDS spectra are collected simultaneously with the backscatter electron images, the image set can be combined to determine the mineral phase present at each location.

### 2.2. Phase II

In this phase, similar polished cementitious composite sections, prepared for Phase I, were studied using BSEM micrographs and EDS analysis to identify microstructural properties, chemical elements and hydration products.

For the purpose of this phase, cement, sand and water, with an addition of only one type of CWA (K type) with two cement types (OPC and PLC) were mixed together to make cubic samples (50 × 50 × 50 mm^3^) of various mixtures of cementitious composite (Table 2). After casting, cubes were plastic wrapped for a period of 24 h, and later, cured in the water bath at 23 ± 2 °C in separate containers to prevent any influence of chemical leachates.

### 2.3. Phase III

Formation of CWA’s crystals was not observed on polished sections of cementitious composite mixtures (neither in Phase I nor in II) due to the conventional harsh surface preparation procedure, including grinding, lapping and polishing, which can abrade crystals from the surface (Figure 2). Hence, fractured surface sections of three CWA mixtures (K, P and X admixtures) were prepared by simply cutting unpolished samples with saw and analyzed them using SEM and X-ray analysis. This phase covers the details about crystals’ development and their differences in chemical elements. Compared to control specimens, all CWA-modified samples visually indicated the presence of pore-blocking crystals growing inside the matrix. While admixture K showed a low sulfur peak in its EDS spectra, both X and P admixtures indicated the presence of ettringite due to a similar high sulfur peak.

To investigate the morphologies of the hydration products or the shapes of the formed crystals, the third phase in this study included three CWA types, called K, P and X admixtures, as summarized in Table 2. These are commercially available products using crystalline waterproofing technology, and their chemical compositions are proprietary to their manufacturers and are not available. In this phase, fracture surfaces of the cementitious composites with and without CWAs were characterized by SEM and EDS analysis. 

In Phase III, specimens were broken into the small pieces with fresh fracture surfaces after 56 days of immersion in water in separate containers for further investigations using SEM. The specimens with approximate dimensions of 2 × 2 cm were randomly selected; covered to minimize chance of carbonation (or contamination) of the surface; left at ambient temperature for 7 days; and later placed into a vacuum desiccator to dry. Next, the surfaces of three samples from each mix were coated with carbon to dissipate excess charge from the specimen during imaging.

## 3. Results and Discussion

### 3.1. Phase I

From a BSEM image of a polished section of concrete, an approximate idea of the concrete porosity, degree of cement hydration and cement type used in the mix can be gained all by looking at the image. With less reliability and longer analysis time, this information can be also obtained by SEM images taken from fractured surface of the specimen. In backscatter mode when the surface is smooth, electrons are collected at a very high angle (close to 180°) and bounced back from a sample in a straight line, making it less sensitive to small topographic features. In contrast, secondary electrons are collected at a low angle in order to enhance topographic features. BSE detector is for collecting scattered primary electrons and SE detector is for collecting the electrons “jumping” out samples. Back scattered electrons have high energy and can deliver a high contrast of different phases. The path length of X-rays generated at depth, which is dependent on the X-ray take-off angle, plays an important role in X-ray travelling time and eventually element peaks in the EDS spectrum [43]. Figure 3 illustrates the change in the average porosity of hardened cement paste with the number of spots analyzed in SEM images for cementitious composite specimens with and without CWA. The average porosity was determined as a percentage of the segmented pore area (black spots in Figure 1) to the total paste area in the generated binary image. The values for paste porosity range from 8.5% to 29.3% for all mixtures. The calculated average porosity from 10 analyzed spots for O, O-K, L and L-K mixtures are 19.44%, 16.59%, 18.72% and 17.76%, respectively. As can be seen in Figure 3, addition of CWA into the mix resulted in approximately 15% (O-K mix) and 9% (L-K mix) lower total average paste porosity in cementitious composite specimens when compared with OPC (control) mixtures. Inclusion of PLC instead of OPC in the mix resulted in 5% reduction in total average porosity. The morphologies of the specimens containing CWA materials showed that the formed crystals contributed significantly to reduce the total porosity, which confirmed the high performance of concrete containing CWA materials in practice. For the formed crystals, the other phase of C–S–H gel reduced the water transport of hardened concrete by partially blocking the pore structure [23,26,49]. This finding is also in agreement with Elsalamawy et al.’s study [23], which examines the available space in the matrix of mortars with CWA. In this study, a reduction of 65 ± 5% in available space width was observed for CWA treated samples [23]. Another study by Zizkova et al. [18] also concluded that a higher CWA content (i.e., 1.5% cement mass) can result in a reduction in porosity exposed to aggressive environments. This improvement in porosity reduction is expected to be even higher in concrete than cementitious composite that was tested in this study due to less interconnection between pores. It also indicates the effect of this admixture in blocking pores/cracks and reducing the matrix’s overall porosity. When two different mixtures (O and O-K which contain OPC cement type and K as the additive or L and L-K containing PLC cement type and K as the admixture) share similar mix properties except one variable (presence of CWA), it can be stated that reduction in the porosity leads to reduction in the permeability. To verify this, water permeability tests were conducted by the authors of [26] in accordance with the DIN 1048-part 5 standard [37]. Only CWA treated mixtures satisfied a maximum penetration depth of 50 mm by having the average values of 40 mm for OPC-CWA and 25.7 mm for PLC-CWA [26]. When compared with OPC group, PLC concrete group indicated 20% and 37.5% lower water penetration depth for un- and CWA-treated, respectively [26]. The results showed significant lower water penetration depth and permeability coefficient for K treated concrete specimens compared to control ones. It should be noted that for the purpose of reducing permeability, all pores do not need to be filled. Blocking only pathway and interconnection of pores can lead to significant reduction in permeability. Hence, around 15% reduction in porosity of CWA treated specimens at micro level can lead to the permeability reduction up to 60% reported in [26]. This agreement indicates the efficiency of K treated samples at both micro and macro levels in terms of reducing pore structures and interconnectivity. As mentioned, using PLC instead of OPC showed a slight reduction in paste porosity. PLC has higher limestone content (~5%) than OPC which results in changes in capillary pores due to several physical effects, such as the dilution effect, the filler effect and heterogenous nucleation [50]. The filler effect suggests a modification in the initial porosity while limestone particles act as nucleation sites leading to increase the early hydration of cement, thereby producing a more disoriented crystallization of CH [50]. Authors could not find any similar studies that report such microstructural features of crystalline waterproofing admixture-treated systems at the micro level using SEM and EDS analyses.

### 3.2. Phase II

To study morphological aspects and hydration products of polished cementitious composite sections, backscatter mode of SEM along with X-ray detector was used. Fracture surfaces from specimen are typically useful to study morphology of the hydration products or the shapes of the clinker mineral crystals while polished sections are generally much more beneficial in evaluating concrete defects [43]. Figure 4 is a comparison of BSEM images of cementitious composite made with the two cement types and also CWA addition (K admixture in this case). Figure 4a,c illustrates two representative BSEM micrographs for control (OPC) and PLC specimens, whereas BSEM images for CWA modified cementitious composite samples (O-K and L-K mixtures) are shown in Figure 4b,d. In these images, various hydration phases were identified by using chemical element maps, taken from EDS analysis, and the degree of image brightness as marked by different colours in the micrograph. The difference in image grey levels is mostly attributed to chemical changes since elements with higher atomic number appear brighter in BSEM image. To distinguish between aggregates and cement paste, those areas, covered by fine aggregates (sand in this case), are marked with “S” sign in orange color. These areas have been removed from original photo during image processing steps as mentioned earlier. In this examination, most of the typical cement hydration products, including calcium silicate hydrate (CSH) and calcium hydroxide (CH), were identified, as marked in Figure 4a,d. The presence of ferrite and aluminate compounds was also observed in the matrix, which has been marked with green boxes for aluminate and pink boxes for ferrite (Figure 4). The cementitious composites with CWA treated system did not indicate any noticeable differences in morphology than those without the admixture and showed similar hydration phases. Cuenca et al. [19] examined samples collected from crack surfaces of two additional concrete specimens, respectively with and without crystalline waterproofing admixture, after three months of immersion in water. They also observed that morphology of hydration products is similar to that typical of an ordinary concrete. In some locations, cracks, shown in yellow boxes in the images, were observed which might have developed during specimen preparation stage because of elevated temperature induced during polishing/grinding procedures. As is clearly visible in Figure 4b,d, some partially hydrated cement particles could be detected as a grey rim of inner hydration products formed around a white core of anhydrous cement. 

Atomic ratio plot is one of the methods to analyze the EDS data and understand more about the hydrated compositions [28]. To examine hydration products of the polished sections, similar procedures, reported in authors’ previous study [28], were followed. The motivation to plot these graphs is also explained in [28]. Figure 5a,b shows atomic ratios plots of Si/Ca vs. Al/Ca and Al/Ca vs. S/Ca (most typical elements exist in the cement after hydration, providing different Ca- and Si-based chemical compounds in the matrix) for more than 20 different spots in each specimen. In ternary diagrams (Figure 5c,d), atomic weights for different cementitious components (e.g., fly ash, cement, metakaolin and slag) were also drawn as a reference to identify any similarity in chemical compositions. It was observed that examined spots in the control PLC mix mostly clustered near CH and CSH regions (Figure 5a), suggesting formation of these main hydration products in examined locations while L-K modified mix showed intermix formation of different hydration products. Similarly, the O-K group showed the same trend as L mix with most points clustered near CH and CSH as can be seen in Figure 5a. Similar to the L-K mix, combination of hydration phases was observed for the control mix. These observations can be also confirmed by Figure 5b. L and O-K mixtures show mostly CSH gel as their hydration product while intermix of mono-carbonate and CSH phases were observed for O and L-K mixtures which indicates some of fine limestone reacted with the cement pore fluid to form mono-carbonate. A considerable reduction in calcium hydroxide content “CH“ was documented in concrete mixture containing CWA materials [23]; this may be due to the presence of highly fineness silica that can react with CH. Presence of CWA in the mix did not indicate any differences in major hydration products than those in control samples. EDS analysis in Cuenca et al.’s [19] study also showed mainly the typical elements of hydration products of cement (calcium, oxygen and silicon in major amounts, in addition to magnesium, aluminum and potassium) formed on the surface of CWA modified specimens which is in good agreement with the findings of this study.

### 3.3. Phase III

#### 3.3.1. Micrographs and Morphological Aspects

In this phase, SEM analysis was carried out on fragments collected from cementitious composite specimens treated with three CWA admixtures in order to obtain useful morphological information and identify formation of any pore-blocking crystals. SEM micrographs of both admixture-treated and control samples, taken from different random locations on the specimen surface is illustrated in Figure 6. Figure 6b–d show respectively the crystals growing in structure-modified samples with K, X and P admixtures, filling cracks/voids while in the control specimens, the cracks/voids did not indicate any sign of crack-filling with crystals (Figure 6a).For better visual comparison, SEM images were sorted in each row from lower (around 3.00–4.00 k) to high (~9.00 k) magnification. These images can be good indicators of admixtures’ performance in filling voids and cracks with crystals which can help in blocking the pathways of aggressive agents to not penetrate deep into the matrix. Other research studies represented similar needle-shaped crystals on fractured surface of CWA treated samples [15,16,19,20,23]. As mentioned earlier, presence of pore-blocking crystals led to decrease in the total porosity of matrix (Figure 3) which can be also beneficial in disconnecting inter-connected matrix pore network, leading to obtain lower permeability values. These crystals, measured from Figure 6b,c, are up to 3–5 µm in length and some of them are less needle-like perhaps signifying either different compositions or geometry development. Ferrara et al.’s study [10] also reported that the particles of the employed CWA showed irregular shape and size in the range of about 1–20 µm; their morphology was similar to that of cement grains. Drochytka et al. [15] also concluded that most of the pores filled with crystals which had a diameter of less than 20 μm.

#### 3.3.2. EDS results

Although differences in nature of these crystals’ and their geometry/morphology might not be obvious from SEM micrographs in Figure 6, the EDS/chemical analysis was conducted to determine the chemistry of crystals and identify their chemical elements. For this phase, the same spots as SEM images taken, were chosen for EDS analysis based on the morphology of the formed crystals and hydrated phases. EDS analysis of K admixture showed mostly typical elements of hydrated control mix (Ca, O and Si in major amounts, in addition to Al, Fe and Mg), illustrated in Figure 7. In this figure, direct visualization and quantification of elements in raw specimen are plotted. High calcium distribution at bottom left corner of Ca elemental mapping indicated that hexagonal structure, observed in the SEM micrograph, is a CH crystal. The EDS microanalysis of Ferrara et al.’s study [10] also highlights the presence of calcium, oxygen, silicon, magnesium, aluminium and potassium in CWA mix. X-ray analysis of the needle-shaped crystals’ regions showed no significant element concentration in the elemental map. It was a combination of different elements in these areas. 

Based on the EDS analysis of needle shaped crystals, the main difference is the sulphur peak. Both P and X have shown signs of sulphur content (Figure 7b,c), but the sulphur peak is not observed in the K needle-like crystal (Figure 7a). On the other hand, ettringite crystals are also known for having a “needle-like” (i.e., acicular) shape. Ettringite’s chemical formula is 3CaO∙Al_2_O_3_∙3CaSO_4_∙32(H_2_O), and is composed of 19.16% Ca, 4.3% Al, 7.66% S and 63% O in atomic percentahes [51]. To differentiate between discovered crystals in treated specimens and ettringite, the specimens were considered for further analyses using EDS analyzer to obtain the chemical elements and EDS spectra of the formed crystals.

From the preliminary analysis, it appears that the composition of the “needle-like” crystals in the CWA specimens with K admixture are different from ettringite due to low peak of sulfur (S) in the EDS graph which is one of the major elements in the ettringite chemical composition. Lower peak of an element in the EDS plot indicates less availability of that element in the matrix where in this case shows that sulfur amount is very low. Hence, needle-shaped crystals observed in the K mix are not ettringite. The EDS spectrum of CWA samples in [10] study was comparable with that of an OPC, except for the peak of sulphur which was slightly higher. This statement aligns with findings of this study which also indicates higher sulphur peak for X and P admixtures while Cuenca et al. [19] confirmed the absence of the sulphur peak in their EDS spectrum. They indicated for the same sample with CWA, the cement matrix was covered with very fine acicular products which were identified as very common microstructures found in self-healed samples; specifically, ettringite crystals were formed inside and filled the crack. Other cementitious composites with different admixture (P and X) types were also investigated to understand the exact chemical compositions of these crystals (ettringite or not). The formation of ettringite or other calcium sulfo-aluminate hydrate when using CWA was dismissed according to the BSEM and XRD results obtained in the [52] study.

The results obtained from the EDS spectra for X and P admixtures indicated that not all admixtures can possibly develop needle-like crystals—acting as a physical membrane—which have different compositions from ettringite. Ettringite’s X-ray spectrum typically has the strongest peaks due to calcium, sulphur and aluminium according to the spectrum plotted in Winter’s book [43]. In admixtures X and P, similar signs of ettringite were spotted in different locations as illustrated in Figure 7b,c. The ettringite crystal holds a lot of water and once it forms, a large increase in solid volume will occur in available pore spaces which leads to expansion of the paste and crack formation particularly around aggregate particles [53]. Hence, the delayed formation of ettringite needs to be prevented inside the cement matrix.

Similar procedures, described for Figure 5, were followed to calculate atomic ratios for particular elements (Figure 8a,b) and draw also ternary diagrams, illustrated in Figure 8c,d. For each cementitious composite type, more than 10 different locations were selected on the specimen where EDS spectra were recorded; then each point was plotted in Figure 8. Among these admixture types, P admixture showed irregular distribution and contained some data points which were located totally outside the specified hydration products regions (Figure 8a,c). X and K admixture treated specimens indicated Si/Ca and Al/Ca ratios approximately between 0.01–0.45 and 0.01–0.23, respectively (Figure 8a). These ratios are within typical range of cement hydration phases [54]. Different phases of calcium silicate hydrate C–S–H have been observed in presence of CWA materials in [23] study. These phases range from round particle to needle like crystals with Ca/Si ratio ranged from 2.4 to 3.2. They also showed intermix compositions of CH, CSH and AFt phases as illustrated in Figure 8a,d. Presence of mono-sulfate (MS) and mono-carbonate (MC) phases in very small ratio were also identified for K and P admixtures (Figure 8b). The MS crystals are only stable in a sulfate deficient solution [55]. In the presence of sulfates, the crystals resort back into ettringite, whose crystals are 2.5 times the size of the MS [55]. It is this increase in size that causes cracking when cement is subjected to sulfate attack [55]. O-P mix indicated considerable amount of MS which shows their higher chance of MS restoration to ettringite. No pure gypsum or AFm phase was detected in the hydration products (Figure 8b). These compounds have significant impacts on setting [56]; early-age strength gain [57]; and longer-term performance factors, such as external sulfate resistance [58,59,60] and the ability to bind radionuclides [61]. Overall, differences were observed between hydration products forming in the presence of CWA, but their differences were not substantial. XRD results from [52] showed that there has not been any significant difference detected between control and CWA concrete samples and in fact, the relative intensity of the CH and ettringite peaks was very similar in all the cases. This finding is in contrary to obtained EDX results in this study which observed differences between chemical elements and morphology presented in different admixtures. They also concluded that the crystalline components of the cement paste were not influenced by the inclusion of CWA; hence, any new crystalline component has not been detected in the CWA concrete, so the formation of a crystalline phase as responsible for the decrease in the water permeability when the addition CWA additive was rejected. They claimed that the main difference detected between the cement matrix of the control concrete and CWA modified one was the formation of the small phases.

## 4. Conclusions

This paper aimed to draw some insights about the microstructural features and durability improvement of cementitious composites modified by adding three different crystalline waterproofing admixtures (CWA) called K, P and X. These CWA types were examined by SEM to confirm previous durability/permeability findings and understand their effects on modifying microstructure, forming pore blocking crystals and hydration phases. Using backscatter-mode SEM images and EDS spectra, examination of polished cementitious composite sections with and without CWA was conducted for the first time. The results showed typical hydration phases, including CSH and CH with different Si/Ca and Al/Ca forming in the control system. Excepting crystal formations in the matrix, CWA-added mixtures indicated noticeably similar morphologies to those without an admixture and showed similar hydration phases. Image processing using ImageJ software indicated a decrease in total average porosity of K treated cementitious composites (about 15% lower) compared tp those with only OPC or PLC cement. Inclusion of PLC in the mix resulted in a slight decrease (around 5%) in average porosity as compared to OPC, while adding CWA into PLC mix led to a ~7% decrease when compared with control PLC mix. It should be noted that the calculated porosity reduction (~15%) in the treated samples is referred to cementitious composite specimens, while in the case of concrete, as previously studied by the authors [26] showed around 60% reduction in concrete water permeability. Hence, slight reduction in total porosity of cementitious composite sample (approximately 15%) could result in a much higher reduction in actual concrete permeability (~60%). 

Observation of crystals using conventional SEM surface preparation is not practical due to possible mechanical damage that occurs through the polishing process. Due to the limitation of conventional SEM surface preparation, the fractured surfaces of cementitious composite specimens were analyzed to investigate the shapes and chemistries of crystals that were generated as a result of adding CWA admixtures. The SEM micrographs taken from specimens’ fractured surfaces showed formation of pore-blocking crystals for all treated mixtures, while similar spots in untreated sections did not indicate such crystal growth. Although needle-shaped crystals were observed in the treated cementitious composite specimens showing similar shape, the chemical analysis of these crystals proves that they have different chemistry and chemical elements. For both P and X treated samples, a peak of sulphur has been observed, which is a sign of ettringite presence in the hydration products, but K sample does not show sulphur peak on EDS spectrum (no sign of ettringite). This study successfully sheds light on the microstructural development (both physical and chemical properties) of cementitious system treated with crystalline waterproofing technology.

## Figures and Tables

**Figure 1 materials-13-01425-f001:**
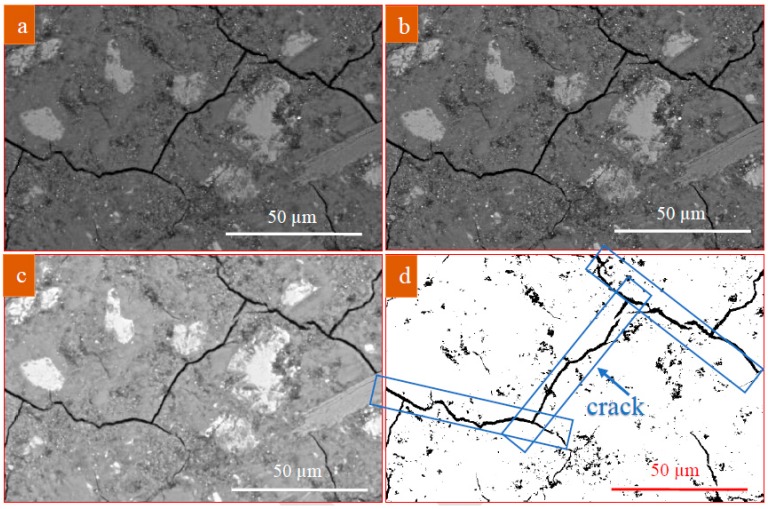
(**a**) A representative SEM micrograph; (**b**) the noise reduced image; (**c**) the adjusted grayscale image; and (**d**) the binary image.

**Figure 2 materials-13-01425-f002:**
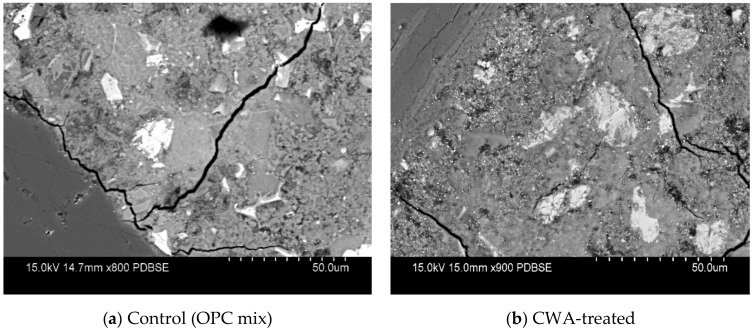
Back-scattered SEM images of (**a**) control and (**b**) CWA-treated samples indicating no crystal presence due to the harsh surface preparation procedure.

**Figure 3 materials-13-01425-f003:**
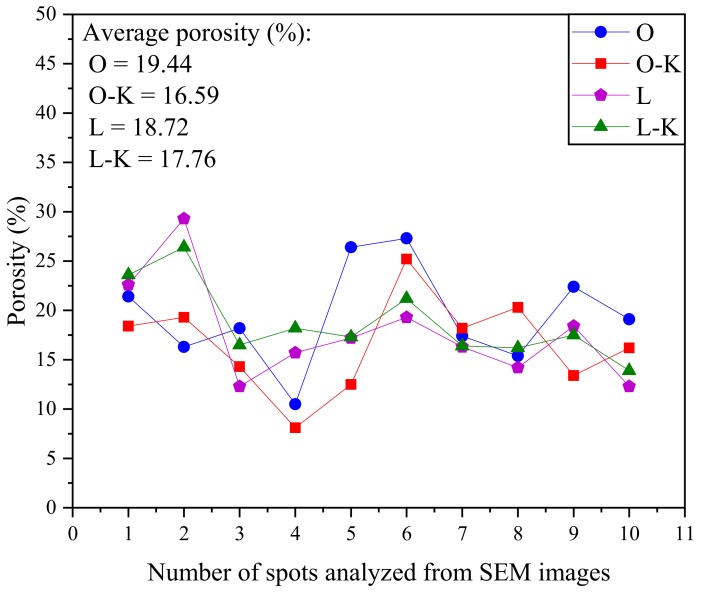
Variation of average cementitious composite porosity with the number of images analyzed with and without CWA.

**Figure 4 materials-13-01425-f004:**
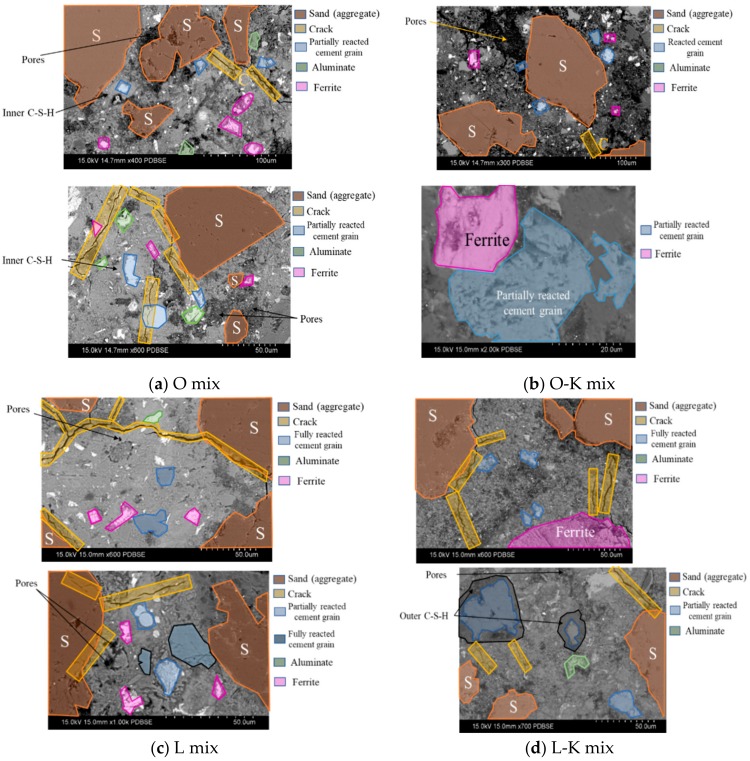
BSEM images of polished cementitious composite sections with and without CWA (**a**) O mix (**b**) O-K mix (**c**) L mix (**d**) L-K mix.

**Figure 5 materials-13-01425-f005:**
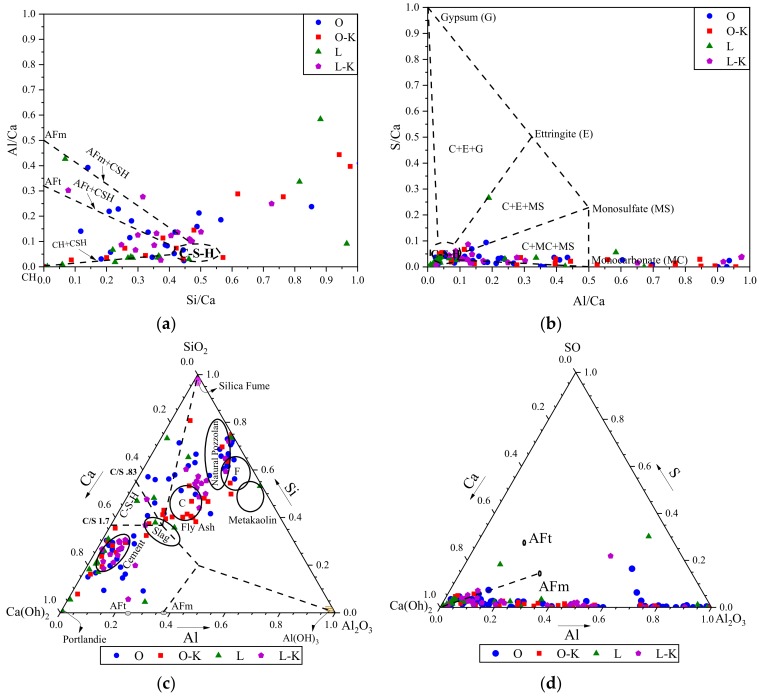
CWA un-/treated cementitious composites—atomic ratios of (**a**) Si/Ca vs. Al/Ca, (**b**) Al/Ca vs. S/Ca; and ternary diagrams of (**c**) Ca-Si-Al, (**d**) Ca-S-Al.

**Figure 6 materials-13-01425-f006:**
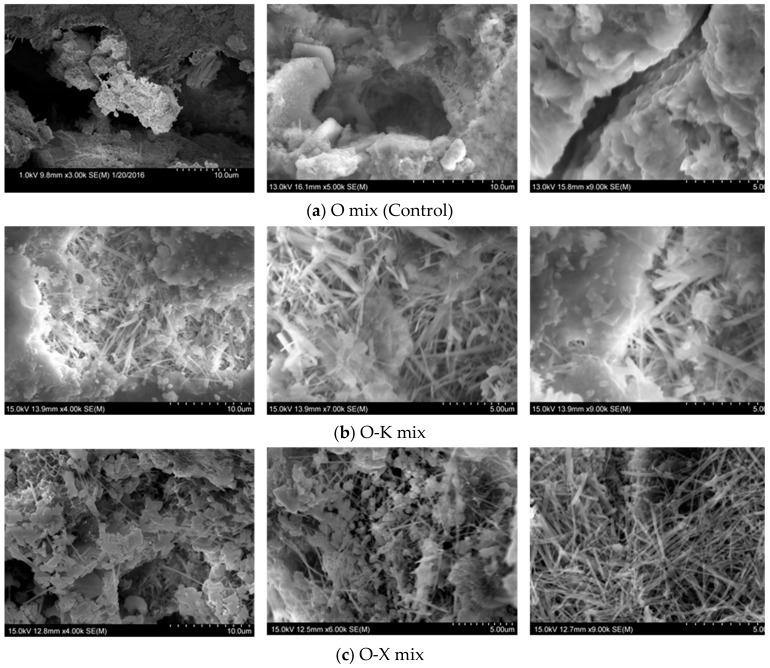
SEM micrographs of cementitious composites with and without CWA (taken from different locations) (**a**) O mix (Control) (**b**) O-K mix (**c**) O-X mix (**d**) O-P mix.

**Figure 7 materials-13-01425-f007:**
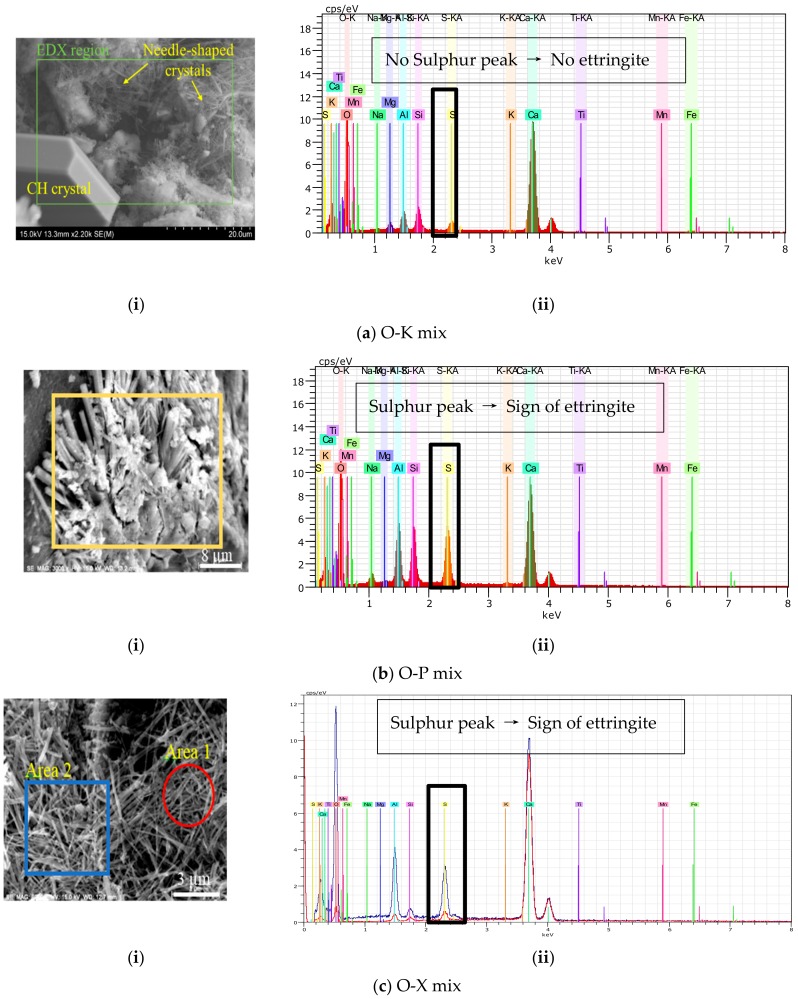
(**i**) SEM micrograph and (**ii**) EDS spectrum of cementitious composite CWA modified mixtures (**a**) O-K mix (**b**) O-P mix (**c**) O-X mix.

**Figure 8 materials-13-01425-f008:**
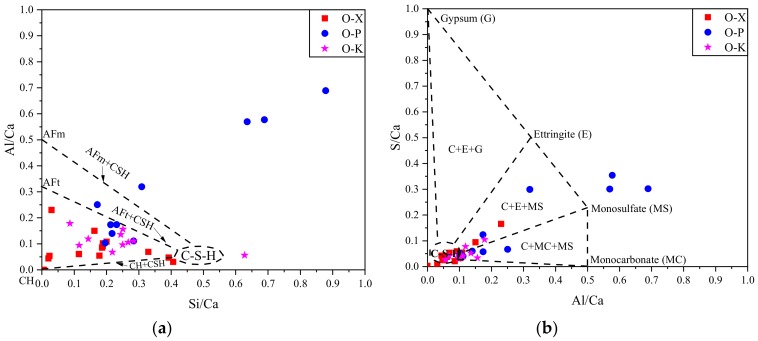
Different CWA treated cementitious composites: atomic ratio plot of (**a**) Si/Ca vs. Al/Ca, (**b**) Al/Ca vs. S/Ca; and ternary diagrams of (**c**) Ca-Si-Al, (**d**) Ca-S-Al.

**Table 1 materials-13-01425-t001:** Chemical and physical properties of crystalline waterproofing admixtures (CWA).

Physical Properties				
	K Admixture	P Admixture	X Admixture	Cement
Color	Gray	Gray	Gray	Gray to gray-brown
Texture	Powder	Powder	Powder	Granular Solid
Particle size	40–150 µm	NR	NR	NR
Specific Gravity	2.6–3.0	NR	2.0–2.8	3.15
pH	12–14	10–13	9.1–9.8	13
Solids	100%	100%	100%	100%
**Chemical Properties**				
Portland Cement	28%–40%	65%–80%	35%–60%	-
Silica Sand	30%–40%	-	30%–40%	-
Alkaline Earth Compounds (Calcium dihydroxide)	-	-	5%–20%	-
Calcium magnesium hydroxide	-	1.5%–6%	-	-
Calcium magnesium hydroxide oxide	-	1.5%–6%	-	-
Calcium hydroxide	5%–20%	1%–2%	-	-
Proprietary Portion	NR	15%–40%	NR	-

**Table 2 materials-13-01425-t002:** Cementitious composite mix design and proportions.

Phase	Mix ID	Materials			Curing Age (Days)
		Mix Proportions (kg/m^3^)			
		Cement	Fine Aggregate	Water	
Phase I and II	O (OPC)	736	2207	368	28
	O-K				
	L (PLC)				
	L-K				
Phase III	O (Control)				56
	O-K				
	O-P				
	O-X				

**O**: Ordinary Portland cement (OPC), **L**: Portland limestone cement (PLC), **O-K**: OPC and K admixture, **L-K**: PLC and K admixture, **O-P**: OPC and P admixture, **O-X**: OPC and X admixture.

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
