# Peer review of "Inventive Microstructural and Durability Investigation of Cementitious Composites Involving Crystalline Waterproofing Admixtures and Portland Limestone Cement"

_materials, 2020, doi:10.3390/ma13061425_

Round 1
Reviewer 1 Report
This paper presents a study about characterization of microstructure and chemical elements of hydration products for cementitious composite with Crystalline Waterproofing Admixtures (CWA). The manuscript is well written and the reading is done fluently. The problem to be solved in the study is well defined in the introductory chapter. The only element that requires improvement is the placement of the words in the keywords: Please start with the general words till the more specific terms.
I believe that the methodology is rationally divided into logical phases of increasing evolution of the observed elements.
The discussion of the results is consistent with what can be concluded from the observed elements.
Finally, in my opinion, the paper can be accepted as it is presented, without prejudice to the improvement of keywords.
Reviewer 2 Report
The paper analysis microstructural and durability of cement-based composites with waterproofing admixtures and Portland limestone cement. The article is nicely written, however, I have some remarks:
1) Introduction section is quite poor. It gives an explanation of what is what but there is no presentation of the results observed or observations made by other authors in the same or similar fields.
2) All the Figures starting from Figure 2 should have the scale bar presented exactly as in Figure 1.
3) Could you please somewhere in the paper explain the difference between K, P and X admixtures?
4) What is (i) and (ii) in Figure 7? (a) part contains them but (b) part does not have them.
5) I missed more discussion of the obtained results with other authors works and observations.
6) Citations in the whole article should be done as shown in journal's article template.
Reviewer 3 Report
The paper presents microstructural and durability investigation of cementitious composites involving crystalline waterproofing admixtures. The effect of such admixtures on the properties of the concrete mixtures is highly interesting and important. However:
1/ The work is written in an incomprehensible and lengthy manner, and therefore it is difficult to understand and connect all the parts of the work.
2/There are no scientific and chemical explanations for the results obtained for the various types of waterproofing admixtures.
3/ The results given for the different images and figures are not well understood and explained. For example, on page 6, line 211 - Are the changes significant and what is the reason for the changes. Is it right to calculate an average for a wide range of results?
4/ Any chemical or physical properties of the cement, admixtures, and aggregates used were presented.
5/ What was the mixing procedure?
6/ The topic of the paper was to investigate the durability. However, nothing was said about the durability of the various mixtures.
7/ What are the chemical differences between the different admixtures?
8/ It is important to show water permeability and other performance results of the concrete mixtures and not to assume that the properties are supposed to be improved.
This topic of the research is very interesting academically and industrially, but it is recommended to reorganize the article, give information on all materials used, and also to provide various properties of the harden concrete mixtures to support the given results on the images analysis.
Round 2
Reviewer 2 Report
Authors have corrected everything according my remarks.
Reviewer 3 Report
The paper was revised according to the comments given.